# Habituation of Central and Electrodermal Responses to an Auditory Two-Stimulus Oddball Paradigm

**DOI:** 10.3390/s24155053

**Published:** 2024-08-04

**Authors:** Gianluca Rho, Alejandro Luis Callara, Enzo Pasquale Scilingo, Alberto Greco, Luca Bonfiglio

**Affiliations:** 1Dipartimento di Ingegneria dell’Informazione, University of Pisa, 56122 Pisa, Italy; gianluca.rho@phd.unipi.it (G.R.); alejandro.callara@unipi.it (A.L.C.); enzo.scilingo@unipi.it (E.P.S.); alberto.greco@unipi.it (A.G.); 2Research Center “E. Piaggio”, University of Pisa, 56122 Pisa, Italy; 3Department of Translational Research and New Technologies in Medicine and Surgery, University of Pisa, 56126 Pisa, Italy; 4Unit of Developmental Neurorehabilitation, Neuroscience Department, Pisa University Hospital, 56126 Pisa, Italy

**Keywords:** orienting response (OR), habituation, recovery, dishabituation, skin conductance response (SCR), electrodermal activity (EDA), oddball paradigm, event-related potentials (ERPs), P3(00)

## Abstract

The orienting reaction (OR) towards a new stimulus is subject to habituation, i.e., progressively attenuates with stimulus repetition. The skin conductance responses (SCRs) are known to represent a reliable measure of OR at the peripheral level. Yet, it is still a matter of debate which of the P3 subcomponents is the most likely to represent the central counterpart of the OR. The aim of the present work was to study habituation, recovery, and dishabituation phenomena intrinsic to a two-stimulus auditory oddball paradigm, one of the most-used paradigms both in research and clinic, by simultaneously recording SCRs and P3 in twenty healthy volunteers. Our findings show that the target stimulus was capable of triggering a more marked OR, as indexed by both SCRs and P3, compared to the standard stimulus, that could be due to its affective saliency and relevance for task completion; the application of temporal principal components analysis (PCA) to the P3 complex allowed us to identify several subcomponents including both early and late P3a (eP3a; lP3a), P3b, novelty P3 (nP3), and both a positive and a negative Slow Wave (+SW; −SW). Particularly, lP3a and P3b subcomponents showed a similar behavior to that observed for SCRs , suggesting them as central counterparts of OR. Finally, the P3 evoked by the first standard stimulus after the target showed a significant dishabituation phenomenon which could represent a sign of the local stimulus change. However, it did not reach a sufficient level to trigger an SCR/OR since it did not represent a salient event in the context of the task.

## 1. Introduction

The P300 component (otherwise known as P3 or Late Positive Complex, LPC) of the event-related potentials (ERPs) and the skin conductance (SC) responses (SCRs) are bound together to the extent that stimuli administered during the experimental paradigm can cause an implicit orientation reaction (OR) of which P3 and SCRs are believed to represent the central and the bodily sympathetic manifestations, respectively [1].

### 1.1. Orienting Reaction

The OR is an automatic behavioral response that redirects attention towards a new and/or significant environmental stimulus, allowing living beings to promptly respond to the stimulus or to enact cognitive exploration behaviors towards the stimulus itself (orientational-investigative behavior), while supporting the collection and memory retention of information about the surrounding world [2].

The repetition of a standard stimulus creates a neuronal representation (i.e., a memory trace) of the stimulus context. Any subsequent incoming stimulus is compared with this representation in working memory: if the current stimulus matches with the memory trace, the representation is maintained; if the current stimulus does not match, a novelty signal is generated which triggers an OR [2] and, at the same time, requires an updating of the neural representation [3,4]. Conversely, a significant stimulus triggers an OR based on a signal of accordance with the memory trace of its biological relevance. If the current stimulus is considered to be both novel and significant, the OR is strengthened [2,3].

The repetition of a stimulus, irrespective of its novelty, causes habituation, i.e., the progressive reduction in the OR magnitude. This is a fundamental learning phenomenon that allows living beings to automatically inhibit responses to harmless or irrelevant environmental stimuli, freeing attentional resources in favor of more important stimuli from a biological adaptive point of view [5]. The significance of a stimulus is able to counteract habituation by slowing it down [2].

### 1.2. OR and SC Responses

The OR is accompanied by an increase in sympathetic autonomic activity (the so-called sympathetic arousal), which regulates homeostatic functions of the organism to behavioral demands through the enhancement of various bodily functions, including increased heart rate, blood pressure, and sweating [1]. The latter determines measurable phasic changes in SC at the surface, known as skin conductance responses (SCRs), which represent a reliable measure of the OR magnitude, even when the behavioral response is not explicitly manifested [2,6,7]. Indeed, SCRs parallel the changes that characterize the OR as a response subject to habituation: reduction in response to repeated stimuli, recovery of response with a change in stimulus (re-orientation or recovery), and increase in response to the resumption of the original stimulus (dishabituation) [5,8].

### 1.3. Oddball Paradigm(s) and ERPs

The most commonly used experimental paradigm to elicit ERPs is the two-stimulus oddball paradigm, where a regular sequence of standard stimuli (*p* = 0.8) is administered with some deviant stimuli (*p* = 0.2) unpredictably embedded. Evoked potentials are extracted by separately averaging EEG epochs related to the standard stimuli and the deviant stimuli, respectively. In this context, the P3 is one of the most studied components. The P3 is a cognitive component related to event evaluation rather than to stimulus sensory perception, and can be evoked within any sensory modality. In the present work, however, we refer to the P3 evoked by means of the auditory tonal oddball paradigm.

In the oddball passive variant, the administered stimuli are not accompanied by any instruction (other than passively listening). Therefore, according to Sokolov’s orientation reaction model [2], the repetition of standard stimuli creates a memory trace of the stimulus context, against which each incoming stimulus is compared. Any subsequent matching stimulus contributes to the creation/maintenance of such a memory trace; on the other hand, deviant (i.e., novel) stimuli generate a mismatch with the established memory trace, producing an implicit OR. In turn, if the novel stimulus is repeated, such an OR undergoes an amplitude decrease due to habituation.

Both the standard and deviant stimuli elicit the exogenous (or sensory) components N1 and P2, which are maximally recorded at the vertex after approximately 100 and 200 ms, respectively. They possess not only exogenous (i.e., sensory) characteristics, since they vary with stimulus physical properties [9,10], but also partly endogenous (i.e., cognitive) characteristics, being modulated by subjective mental operations, such as attention to tones [11]. On the one hand, the N1 seems to reflect either the result of a sort of attention-triggering process which represents a first readout of information by the primary auditory cortices or even the formation of the stimulus memory trace that will be used in subsequent comparison processes. The N1 can indicate the arrival of potentially discriminable information at the auditory cortex, but it is not sufficient by itself to confirm stimulus discrimination [12]. On the other hand, the P2 reflects a subsequent stage towards stimulus discrimination [13,14]. It is still a matter of debate whether both of these components are partially subject to habituation or not at all [15].

Only deviant stimuli, however, additionally evoke the so-called endogenous (or cognitive) components, namely MMN/N2a and P3a. The first one is recorded at frontal and central regions of the scalp after 100 to 250 ms from stimulus onset [16]. It is believed to be generated by an automatic pre-attentive process that detects a mismatch between the incoming deviant stimulus and the sensory memory trace (i.e., echoic memory) of the auditory regularity of the previous stimulation. The presence of the MMN indicates that the change-detection system of the auditory cortex discriminates the standard/deviant discordance, but at a mostly unconscious level. The amplitude and latency of the MMN depend on how much the deviating stimuli differ from the standards [12,17].

The P3a is distributed over the fronto-central regions of the scalp with maximal amplitude over midline sites and latency between 250 and 300 ms from the stimulus onset [16]. Generators have been found within the anterior cingulate, superior temporal, frontal, and parietal cortices [18,19]. It would reflect either the fact that the stimulus was noticed by some attention-triggering mechanism or, perhaps, a redirection of attention monitoring (i.e., an attentional switch) [12]. It is a matter of discussion whether it reflects an attention change which is triggered top-down (voluntary) or rather bottom-up by deviant stimuli, without necessarily indicating the conscious perception of the change [12].

In the active form of the oddball paradigm (otherwise known as the no-go/go paradigm), the participant is instructed to ignore (no-go) standard stimuli and to perform a behavioral response (go) whenever a deviant stimulus (i.e., target) is presented. The response can be either explicit, such as pressing a button in response to each target stimulus, or implicit, such as mentally counting target stimuli as they occur. In the active oddball, according to the Sokolov’s orientation reaction model [2], the deviant stimulus possesses not only the property of a novel stimulus, as it conflicts with the memory trace formed from the repetition of standards, but it also has the property of a significant stimulus, as it matches with the memory trace built by task instructions reception which also makes the deviant stimulus a task-relevant stimulus (i.e., target). In fact, if the current stimulus, in addition to being novel, is also significant, the OR is strengthened [2,3], and the effects of habituation possibly slowed down.

The components recorded with these task instructions are the N2b and the P3b (also simply called P3 or P300). N2b has maximum amplitude at frontal and central sites, and peaks between 245 and 290 ms after stimulus onset [20]. N2b probably represents the pre-conscious perceptual registration of the stimulus change and perhaps the early stages of stimulus evaluation and classification [12]. P3b has a centro-parietal distribution that is maximal over midline scalp sites and peaks at 300 ms or more after the stimulus onset [16]. Generators have been found in the temporo-parietal junction (namely neighboring areas to the anterior intraparietal sulcus and superior temporal sulcus), medial temporal lobe (namely hippocampus and amygdala), lateral and medial prefrontal cortices, and insula [21,22,23]. A recent study has also identified generators in correspondence with premotor and motor areas, suggesting a link between the P3b and the action (implicit or explicit) required by the task [23]. The P3b would be recorded when an update of the stimulus context representation (according with Sokolov’s orientation reaction model) is required, that is, when an incoming stimulus (in this case, a deviant, target stimulus) does not match with it. According to some authors, it would indicate the conscious detection of the target stimulus (i.e., the recognition of the incoming stimulus as a target stimulus) [12,17], but the question is still very debated [24]. Yet, it is worth noting that the hypotheses regarding the mental processes underlying the genesis of the P300 are manifold, including priming (i.e., violations of primed dispositions about what to perceive due to the frequency of stimuli occurrence), cognition, memory storage, context updating, closure of cognitive epochs, response facilitation, decision-making, reactivation of stimulus-response links, and production of conscious representations (for a critical review see [24]).

Further variants are represented by the third-stimulus oddball and the novelty oddball. In the former, a third tone without previous instructions is added, whereas in the latter, an environmental stimulus is introduced. In both cases, the additional stimulus functions as a distractor. The recorded components are the novelty N2 and the novelty P3 (nP3), previously believed to be superimposed to MMN/N2a and P3a, respectively [3,25].

The prevailing opinion, at present, is that any P300 represents the result of the temporal overlapping of different subcomponents (the so-called P3s family), which would preferentially emerge as distinct entities (assuming peculiar characteristics of latency, topographical distribution on the scalp, localization of the cortical generators, and functional meaning) depending on the different characteristics of the task [25,26].

### 1.4. P300 and OR

The separation of P3 subcomponents may be carried out not only a priori, by means of specific oddball variants, but also a posteriori, using complex analysis methods such as principal component analysis (PCA). The latter methodology allows us to identify individual components based on their contribution to the global variance of data. However, while on the one hand the subcomponents separation allows to assign a precise functional meaning to each of them (albeit sometimes putative), on the other hand, it makes it more complicated to assign to P3 the qualification of OR neuronal correlate. In fact, none of the subcomponents of the P3, taken individually, has ever shown the same ability as the P3 as a whole (i.e., LPC) to reflect the SCRs dynamics in studies concerning habituation, recovery, and dishabituation of the OR [27].

### 1.5. Intrinsic Habituation to the Oddball Paradigm

The oddball paradigm is characterized by the repetition of stimuli (standard and target), which undoubtedly allows the ERPs to be extrapolated from the background EEG oscillations, but at the same time, exposes the ERPs (and SCRs) to the effects of habituation. Furthermore, the stimulus presentation sequence is characterized by the occasional occurrence of a stimulus that deviates (target) from the original stimulus (standard) and the subsequent repetition of the original stimulus (standard) after the deviant one (target).

Therefore, despite not having been specifically designed for this purpose, the two-stimulus oddball paradigm would already possess features that should allow us to estimate the extent of the intrinsic phenomena of habituation, recovery, and dishabituation that ERPs are subjected to, due to the fact of having been evoked by this type of paradigm. Given the widespread diffusion of the oddball paradigm in clinical and experimental psychophysiology, this aspect seems to be of paramount importance.

To also investigate the aforementioned effects on SCRs, which are characterized by slower and more delayed dynamics compared to ERPs, we adapted the standard oddball paradigm by using longer stimulus onset asynchronies (SOAs) than usual (i.e., between 4 and 5 s). This allowed us the resolution of the individual SCRs without causing their temporal summation.

### 1.6. Aims and Scope

The aim of the present work was to unveil any correspondence between SCRs (as typical peripheral OR measures) and P3 subcomponents (as potential central OR candidates) by studying habituation, recovery, and dishabituation during a traditional two-stimulus auditory oddball paradigm. In particular, we investigated whether the amplitude of SCRs and P3 subcomponents (a) decreased with stimulus repetition (habituation), for both standard and target stimuli, (b) increased with stimulus change (recovery), and (c) increased with the resumption of the original standard stimulus after the change (dishabituation). Notably, habituation of SCRs was investigated in terms of both their amplitude and frequency of occurrence, where the latter is expressed as the percentage of responses with respect to the total number of stimuli for each of the two stimulus categories.

## 2. Materials and Methods

### 2.1. Experimental Protocol

Twenty healthy volunteers (age 24 ± 4, 10 females) participated in this study. This study was conducted in accordance with the guidelines of the Declaration of Helsinki, and all participants gave written informed consent to participate.

Participants were conducted into a quiet room and were asked to sit in a comfortable chair. They were asked to keep their eyes closed and minimize movements for the entire duration of the experiment. The experimental protocol consisted of 3 min of initial rest, followed by an auditory oddball task. The task consisted of three consecutive blocks of 125 auditory stimuli each, with 80% standard stimuli (frequency: 1.0 kHz) and 20% target stimuli (frequency: 1.5 kHz). All stimuli were monaural tones with a duration of 200 ms, rise and fall times of 5 msec, and an intensity of 60 dB SPL. The presentation of the stimuli (i.e., standard/target) was pseudo-random, meaning that it was possible to present two target stimuli in a row. SOAs were randomly chosen between 4 s and 5 s to allow for a reasonable recovery of stimulus-specific (i.e., evoked) SCRs. Stimulation blocks were interleaved by 1 min of rest. Participants were instructed to focus on the occurrence of target stimuli and to count them across the three blocks to guess their correct number.

### 2.2. EEG and SC Acquisition

EEG and SC were acquired synchronously using a DSI-24 system from Wearable Sensing at the sampling frequency of 300 Hz. The headset was composed of 19 dry electrodes placed according to the 10–20 international system. Electrode impedance was always kept below 1 MΩ in accordance with the manufacturer’s guidelines. All channels were referenced to Pz. SC was recorded via a dedicated auxiliary channel in the EEG cap, applying an exciting voltage of 0.5 V. A pair of Ag/AgCl electrodes were placed on the proximal phalanx of the first and second fingers of the non-dominant hand, respectively. Proximal sites were chosen to minimize the presence of movement artifacts in the recordings [28].

### 2.3. SC Signal Processing

SC can be modeled as the superimposition of two components: a slow-varying tonic component and a fast-varying phasic component [28]. The latter is of particular interest for this study, as it contains information about SCRs.

A well-known issue in the analysis of SC concerns the time interval occurring between two consecutive stimuli. More specifically, due to its slow dynamics, a time range of 10–20 s is necessary for a single SCR to fully recover its baseline. However, inter-stimulus intervals commonly adopted in cognitive and neuroscientific experimental protocols—as those implemented in this paper—are shorter (i.e., 1–5 s), thus leading to the overlap between consecutive SCRs. Hence, identifying individual evoked responses and relating them to their triggering auditory stimulus becomes tricky. A potential solution to this problem concerns the estimation of the underlying sudomotor nerve activity (SMNA). SMNA represents the sparse and discrete bursts of the sympathetic afferent sudomotor fibers which drive the generation of the observed SCRs. These bursts are characterized by a higher temporal resolution compared to SCR activity and can thus be exploited to identify the time instants at which peripheral sympathetic responses evoked by auditory stimuli have occurred [29,30]. In this light, we followed the procedure implemented in [29] to estimate the SMNA and exploit its time resolution to identify the stimuli that elicited a peripheral sympathetic response. Such an approach is based on the convex-optimization-based cvxEDA [30] model, which provides an estimation of the SMNA from observed SC responses. Notably, we hypothesized that peripheral sympathetic responses evoked by auditory stimuli can be precisely identified over time through nonzero SMNA bursts.

Operationally, for each subject, we downsampled the SC to the sampling frequency of 50 Hz, and we performed a Z-scoring on the data. Then, we estimated the SMNA through cvxEDA. We set the sparsity parameter of the model to $8\times 10^{-3}$ as a trade-off between noise suppression and distortion of the solution. Indeed, larger values of this parameter yield a higher sparsity of SMNA responses and thus a stronger suppression of spurious spikes, but also more attenuation of true physiological ones. On the other hand, smaller values yield a less distorted but noisier solution. We set all the other parameters of the model at their default recommended value [30]. For each standard and target stimulus, we extracted epochs from both the phasic component of SC and the SMNA. These epochs started at the time onset of each stimulus administration and lasted 5 s. We assumed that stimuli evoking an OR were associated with nonzero SMNA bursts occurring 1–5 s after their presentation. Such a time interval is in line with several studies indicating that a stimulus-evoked SCR is observed to occur within that range of latency after stimulus onset [28,31]. Accordingly, within each epoch, we extracted the latency and amplitude of the first SMNA peak occurring later than 1 s from the stimulus onset. The procedure is schematically illustrated in Figure 1.

For each stimulation block and for each auditory stimulus (i.e., standard, target), we then extracted the average amplitude of SCRs, and the percentage of SCR occurrence observed in response to the stimulus. Moreover, to investigate the dishabituation of peripheral sympathetic responses to standard stimuli, we extracted the average amplitude of SCRs evoked by the standard stimuli occurring immediately before (pre-target) and immediately after (post-target) the presentation of targets, respectively.

### 2.4. EEG Signal Processing

We preprocessed the EEG signal using EEGLAB [32]. We filtered the data with a zero-phase lowpass antialiasing filter and then we performed a downsampling to the sampling frequency of 100 Hz. Next, we applied a zero-phase high-pass filter at the cutoff frequency of 0.5 Hz to improve data stationarity for subsequent processing steps. We removed flat and poorly correlated channels by exploiting the method presented in [33]. Specifically, the correlation coefficient was computed between each channel and its reconstruction based on the spherical interpolation of its neighbors. Channels were removed if their correlation coefficient was less than 0.8. We recovered the removed channels using spherical interpolation, and we re-referenced the data to the earlobes. We decomposed EEG data through independent component analysis (ICA) [34], and we removed independent components associated with artifact activity (e.g., muscles, eye movements) through visual inspection of their associated time course, scalp map, and power spectrum. We epoched the EEG data in the −200, 1000 ms range with respect to each stimulus onset (i.e., 0 ms), and after a visual inspection, we removed epochs containing residual artifact activity. An average number of 295 epochs (minimum: 280; maximum: 300) and 74 epochs (minimum: 70; maximum: 75) was retained over the subjects for the standard and target stimuli, respectively. We performed baseline correction of clean epochs by subtracting the average potential in the pre-stimulus interval (i.e., from −200 ms to 0 ms), and we estimated subject-average ERPs by grouping epochs in accordance with the stimulus type (standard/target) and stimulation blocks, for a total of six conditions (standard/target stimuli over three blocks). Moreover, for each subject, we averaged together EEG epochs associated with pre-target and post-target standard stimuli. This averaging was performed with the aim of evaluating the presence of dishabituation of ERP responses to standard stimuli.

### 2.5. Temporal PCA

We applied temporal PCA to investigate the occurrence of habituation to target stimuli on individual ERP components. Temporal PCA decomposes ERP responses into a set of orthogonal (i.e., uncorrelated) components, whose temporal course and spatial distribution are described by factor loadings and factor scores, respectively. Factor loadings represent factors’ (i.e., components) time course, and are fixed across channels, conditions, and subjects. Instead, factor scores represent how much a factor contributes to the amplitude (i.e., voltage) of each observation [35]. Operationally, we further epoched subject-average ERPs to target stimuli in the 90–470 ms time range for each stimulation block, and we applied temporal PCA decomposition through the erpPCA functions implemented by Kayser J and Tenke CE (http://psychophysiology.cpmc.columbia.edu/software/; access date: 1 August 2024) [36]. The input data matrix consisted of 1140 trials (i.e., channels × blocks × subjects) by 39 time points. Prior to svd decomposition, the covariance matrix underwent Kaiser normalization. Factors were rotated through unrestricted Varimax rotation. Following their extraction order, we identified factors in terms of their peak latency, polarity, and scalp map. Finally, for each component contributing to the LPC, we extracted the corresponding peak’s amplitude factor scores for each subject and for each stimulation block. Particularly, we selected channels of major interest based on the maximums of the scalp topography (i.e., factor scores) associated with each component [35].

### 2.6. SC Statistical Analysis

We investigated for a possible significant effect of the stimulus type (standard, target) on the amplitude of subject-average SCRs in the time range from 1 s to 5 s after stimulus onset through a permutation-based *t*-test (10,000 permutations, significance level α = 0.05). False positives due to multiple comparison testing were controlled through the false-discovery-rate (FDR) approach [37]. We then investigated for a possible significant effect of the stimulus’ type on the latencies of SCRs through a *t*-test (α = 0.05).

Furthermore, we tested for the occurrence of habituation to the stimuli over time. More specifically, we hypothesized that the amplitude of stimulus-evoked SCRs could decrease as a function of the experimental blocks within which stimuli were presented. Accordingly, we conducted a two-way repeated-measures ANOVA (10,000 permutations, α = 0.05) on the amplitude of SCRs, with stimulus type and experimental blocks as main factors, against the null hypothesis of no significant difference among blocks for any of the stimulus types. We controlled multiple comparisons with the FDR approach. Post hoc comparisons were conducted with multiple *t*-tests, and *p*-values were adjusted with the Bonferroni correction. The same statistical procedure was conducted on the percentage of observed SCRs, computed as the number of responses evoked by a given stimulus type over its total number of presentations within a given block. Here, we hypothesized that habituation could manifest as a reduction in the number of the stimulus-evoked responses over experimental blocks.

Finally, we tested for the presence of dishabituation to the standard stimulus (i.e., an increase in the SCR response with the resumption of the original standard stimulus) over blocks through a permutation-based *t*-test (10,000 permutations, corrected with FDR, α = 0.05) on the amplitude of SCRs observed immediately before and after target stimuli, respectively.

### 2.7. EEG Statistical Analysis

We evaluated significant differences in the average amplitude of each ERP component in response to standard and target auditory stimuli through a permutation-based *t*-test (10,000 permutations, α = 0.05). We focused our analyses on Fz, Cz, and Pz channels, as these channels are widely reported in the literature as key sites where auditory oddball components show their maximum peak of activity [38]. We controlled false positives due to multiple tests with FDR.

We then investigated for the occurrence of habituation to target stimuli over time. To this aim, we performed a permutation-based one-way repeated-measures ANOVA (10,000 permutations, α = 0.05, corrected with FDR) on the factor scores of each LPC component identified through temporal PCA (see Section 2.4), with the three stimulation blocks as the effect of interest. We conducted post hoc comparisons through multiple paired *t*-tests, and we adjusted *p*-values with the Bonferroni correction.

Finally, we investigated for differences in the amplitude of ERPs in response to pre-target and post-target standard stimuli through permutation-based *t*-tests (10,000 permutations, α = 0.05). Since we did not have any a priori hypothesis on the potential differences between the two conditions, we conducted a statistical test for each channel. Multiple comparisons were controlled with FDR.

## 3. Results

### 3.1. SC Statistical Analysis Results

Figure 2 shows the results of the comparison between the average SCR amplitudes in response to standard and target stimuli. As highlighted by the gray shaded area, we observed a significant difference between responses in the time range from 2.8 s to 5.4 s, with SCRs evoked by target stimuli having a significantly greater amplitude compared to SCRs evoked by standards (*p* < 0.05). Of note, these SCRs are obtained from the phasic component estimated by cvxEDA. Accordingly, no confounding effects due to the tonic component’s activity on the reported results are present.

In Figure 3, we report the results of the statistical analysis on the latency of SCRs to standard and target stimuli, evaluated as the first nonzero burst of SMNA activity in the 1–5 s time window after stimulus onset. We did not find any significant difference in the latency of the responses.

A more in-depth analysis of the frequency of SCRs elicited by standard and target stimuli over the three experimental blocks highlighted a significant main effect for both the stimulus type ($F_{1,20}=10.53$, *p* < 0.01) and the stimulation blocks ($F_{2,40}=16.57$, *p* < 0.001) (Figure 4), and a significant effect for the interaction between stimulus type and blocks ($F_{2,40}=29.43$, *p* < 0.001) (Figure 5). Specifically, the percentage of target stimuli that elicited an SCR was higher than the percentage associated with standard stimuli (%SCRtarget = 32%, %SCRstandard = 16%), irrespective of the stimulation block (Figure 4a). Moreover, the percentage of stimulus-evoked SCRs within blocks, irrespective of the stimulus type, showed a significant decrease between consecutive blocks, as shown by the post hoc analysis of Figure 4b (*p* < 0.001). From the post hoc analysis of the interaction between stimulus type and experimental blocks, we further observed that habituation occurred in response to the target stimuli. Indeed, as depicted in Figure 5, there was no significant difference in the number of SCRs elicited by standard stimuli over blocks, whereas the number of SCRs elicited by targets decreased significantly over time. Of note, SCRs elicited by targets were significantly more numerous than those elicited by standards within both the first and second blocks of stimulation. Conversely, within the third block, the SCRs to targets were less than those elicited by standards.

Regarding the statistical analysis of SCR amplitude, we observed a significant effect for both the stimulus type ($F_{1,20}=6.27$, *p* < 0.05) and the experimental blocks ($F_{2,40}=3.62$, *p* < 0.05). On the other hand, we did not find any significant effect for the interaction between the two factors ($F_{2,40}=0.41$, *p* > 0.05). Irrespective of the experimental blocks, the amplitude of SCRs was higher in response to target stimuli compared to those elicited by standard stimuli (Figure 6a). This result is in line with the amplitude analysis of average SCR amplitudes reported in Figure 2. Furthermore, the SCRs’ amplitude showed a significant decrease over time, with a lower amplitude in the second and third stimulation blocks, with respect to the first one (Figure 6b). We did not observe a significant difference between the SCR amplitude of the second and third blocks.

Finally, Figure 7 shows the comparison between the average SCRs elicited by pre-target and post-target standard stimuli. Although the grand averages show a higher amplitude for the SCR elicited by post-targets, with respect to pre-target SCRs, such a difference was not statistically significant probably due to the higher variability of post-target responses (see also Figure 2).

### 3.2. Temporal PCA Results

Figure 8 shows the results of temporal PCA decomposition on the subject-average responses to target stimuli. The first nine factors accounted for about 94.8% of the total variance, with a maximum of 38.7% for the first factor, and a minimum of 1.15% for the ninth factor. We did not consider any further factors since they explained a variance lower than 1%. We associated each of the selected factors to established components in the time range from the N1 to the SW, based on their peak latency, polarity, and scalp topography. We found two subcomponents of the N1, i.e., N1-3 and the N1-1, at a latency of 100 ms and 130 ms, respectively, [10,39], followed by a P2 component peaking at 150 ms and centered at the vertex Cz. Concerning the LPC, we found both a eP3a and a lP3a, peaking at 200 ms and 250 ms, respectively, [40]. The eP3a was distributed towards central and frontal regions around Cz and Fz, whereas lP3a was shifted towards central and parietal regions around Cz and Pz. Furthermore, we observed a P3b component with latency of 310 ms and a topography focused around Pz, followed by a frontal nP3 peaking at 340 ms around Fz [39,40]. Finally, we found two subcomponents of the SW, namely the positive SW (+SW) at 370 ms and the negative SW (−SW) at 430 ms [39,41]. Both +SW and −SW topographies showed a maximum and a minimum, respectively, in the frontal region around Fz.

### 3.3. EEG Statistical Analysis Results

Figure 9 shows the results of the statistical analysis between ERP responses to standard and target stimuli evaluated at Fz (Figure 9a), Cz (Figure 9b), and Pz (Figure 9c), respectively. For each plot, we report the grand-average ERPs obtained from both responses to targets and standards together with their point-wise difference wave. In addition to N1 and P2 components, clearly visible on all three electrodes taken into consideration, N2a and P3a (target wave) as well as MMN (difference wave) could be identified at Fz, while N2b and P3b (target wave) together with sP3 (standard wave) could be identified at Pz. The characteristic progressive increase in latency and amplitude of the P3s towards posterior regions of the scalp was also observed, as well as the separation of P3a (earlier peak) and P3b (later peak) at Pz. Significant differences between standards and targets are highlighted by the gray shaded areas. As expected, we observed a higher amplitude of ERP components evoked by target stimuli compared to standard stimuli, with the exception of the P2 component in both Cz and Pz.

The results of habituation analysis on LPC component amplitude (i.e., factor scores) over stimulation blocks are resumed in Table 1. We observed a significant effect for the stimulation blocks on the amplitude of lP3a component at Cz ($F_{2,40}=4.17$, *p* < 0.05), and for the amplitude of P3b component at Pz ($F_{2,40}=5.02$, *p* < 0.05). Post hoc analysis highlighted a decrease in the amplitude of responses to target stimuli over time, with both lP3a and P3b showing a higher amplitude in the first stimulation block, with respect to the third. As depicted in Figure 8, the grand-average scalp distribution of −SW showed a similar trend at Fz, but without any statistical significance. Likewise, we did not find any significant effect for the stimulation blocks on the factor scores of eP3a, nP3, and +SW.

In Figure 10 and Figure 11, we report the results of the analysis on the differences between ERP responses to pre-target and post-target standard, as well as their grand average scalp distribution for the N1, P2, and standard P3 (sP3) components. We found significant differences between conditions at frontal sites (Fp1, Fp2, Fz), central sites (C3, C4, Cz), parietal sites (P3, P4), and temporal sites (T3). More specifically, post-target responses showed a more negative amplitude in the N1 time range across all significant channels, with respect to pre-target responses. Moreover, post-targets showed a more positive amplitude in the time range of the sP3 component, with respect to pre-targets.

## 4. Discussion

In this work, we showed that both the amplitude and occurrence frequency of SCRs evoked by target stimuli were larger than those evoked by standards. Furthermore, the SCR amplitudes displayed a rapid habituation (i.e., between the 1st and 2nd stimulation block) with repetition of both stimulus categories, while the occurrence frequency manifested a slower habituation (i.e., between the 1st and 3rd block) only for targets.

As expected by the nature of ERPs, the amplitude of the P3 component evoked by targets was greater than that evoked by standards, confirming its recovery with stimulus change.

The consistency shown by SCRs and P3 regarding the higher amplitude of responses evoked by target stimuli seems to suggest that the P3 expresses, in addition to the processing of the stimulus information content (information processing), the processing of its emotional content (emotional processing). In the latter case, the target stimulus would also possess an emotional-affective value due to its own task relevance. Therefore, the target stimulus would not be a completely neutral stimulus from an emotional-affective point of view.

Furthermore, analyses on PCA-derived P3 subcomponents made it possible to highlight that the only ones to show a comparable habituation behavior to that of SCRs were lP3a and P3b, whose amplitude showed a significant reduction across the three blocks of target stimuli.

Finally, it was possible to highlight an amplitude increase in the sP3 upon resumption of the standard stimulation immediately after the target (dishabituation), which, however, was not accompanied by similar behavior of the SCR.

### 4.1. Decrease with Stimulus Repetition of Both SCRs Amplitude and Frequency

The amplitude of SCRs evoked by targets was significantly larger than that of SCRs evoked by standards. Moreover, amplitude decreased with stimuli repetition (habituation), regardless of the stimulus type. Amplitude decrease reached statistical significance between the 1st and 2nd blocks, configuring a rapid habituation phenomenon which could already be considered exhausted by the end of the 2nd block.

The occurrence frequency of SCRs evoked by targets was significantly higher than that evoked by standards (i.e., it was more probable that an SCR occurred in response to a target than to a standard). Moreover, its decrease with stimuli repetition (habituation) was found to be slower for targets than standards, and persisted until the end of the 3rd stimulation block.

Therefore, the two decreasing trends of SCR amplitude and SCR frequency with stimulus repetition appeared to be discordant with each other. The former was relatively fast and independent of the type of stimulus used, while the latter was relatively slow and only limited to the SCRs evoked by targets. We could suggest that these phenomena are governed by two distinct mechanisms: amplitude decrease could deal with response modulation, whereas frequency decrease could deal with the genesis itself of the response (according to an all-or-none mechanism based on a threshold system) [28]. Our interpretation of this latter aspect is that the drive to the decrease in SCR frequency with task-relevant target repetition could be counteracted (i.e., slowed down) by the need to allocate increasing attentional resources to maintain a satisfactory level of task performance. In fact, if on the one hand, the emotional-affective component of ORs towards initial stimuli could at first support (with a bottom-up mechanism) the behavioral response to the task, on the other hand, a greater attentional effort (with a top-down mechanism) may be necessary to maintain performance until the end of the task due to stimulus repetition and the establishment of habituation effects. In this perspective, the OR could become functionally counterproductive to the completion of responses as the task progresses, and being then progressively inhibited.

Concerning the lack of habituation of SCR occurrence frequency to standards, it must be noted that the absolute number of standards per block was 5 times higher than that of targets, even if the percentage of SCRs with respect to the total number of standards turned out to be lower with respect to targets. This element, together with the fact that the standard stimulus is irrelevant to the completion of the task, could have sped up habituation to the point of being largely exhausted by the completion of the 1st block. Thus, the absence of significant changes across blocks could be due to the exhaustion of the steep phase of the decreasing curve occurring with the very initial stimuli of the 1st block. This rapid plateauing masked habituation effects when calculating the average value per block.

### 4.2. P3 Amplitude Decrease with Stimulus Repetition and Recovery with the Original Stimulus Reintroduction

The amplitude of P3 evoked by targets was greater than that evoked by standards. This was expected since ERPs, as endogenous cognitive potentials, are affected by task-relevance rather than physical features of the stimulus. At the same time, however, this is not inconsistent with the idea that it may also represent the response recovery induced by the change played by target stimuli within the invariant sequence of standards. Moreover, the amplitude of both PCA-derived lP3a (at Cz) and P3b (at Pz) significantly decreased with stimulus repetition across stimulation blocks. It is worth noting that statistical significance was achieved precisely in the scalp sites that distinguish the distribution patterns of P3a from that of P3b [3,4,42].

The behavior of these P3 subcomponents with stimulus repetition substantially parallels that of the SCRs (even if with a less steep habituation slope), showing that they can both be considered habituation phenomena that are intrinsic to the classical two-stimulus oddball paradigm.

The present results only partially coincide with those of Barry and colleagues [27], who found correspondence with SCRs for both P3b and nP3, but not for P3a [18,43]. Such a difference between results was somewhat expected since it is known that the proportion of the different subcomponents of the LPC can be profoundly influenced by different paradigms [39,40]. In fact, the paradigm we employed, in addition to not being specifically designed for the study of habituation, also differed in response requirements (count vs. no response), in the order of presentation of the deviant stimulus (variable vs. fixed), in the duration of the SOA (4–5 s vs. 5–7 s) and in the type of analysis performed (block vs. single trial). Nonetheless, the present study allowed us to demonstrate that the PCA-derived P3 subcomponents identified in a classical two-stimulus oddball paradigm are substantially equivalent to that derived from single trial ERPs. This could pave the way for interesting developments in the study of this topic.
On the contrary, our results regarding P3a agree with other authors [3,7,44,45] who consider this subcomponent a central counterpart of OR.

We hypothesize that the slower habituation of lP3a and P3b, compared to that of sP3, might deal with the saliency of significant stimuli, due to their relevance for the individual adaptive behavior [10,46], and/or with the need to maintain the attentional trace of relevant stimuli over time in order to guarantee the achievement of task goals despite the advancing of habituation [10,47,48,49,50]. Indeed, the fact that some P3 subcomponents (i.e., the underlying activation of the cognitive system) habituate more slowly than SCRs (i.e., the underlying activation of the emotional system), could involve the need for maintaining task goals throughout the completion of the task [51]. Therefore, the prevailing activation of the cognitive system over the emotional one should be considered in the perspective of effortfully maintaining the focus of attention on the task. We speculate that the cognitive system could have a somewhat inhibitory effect on the emotional system (then, fostering its habituation) by reducing both frequency and amplitude of ORs as time passes and the number of stimuli increases, in order to optimize both efficiency and energetic sustainability of the performance.

### 4.3. Dishabituation of ERPs with Stimulus Resumption

The amplitude of the standard P3 (sP3) [52] obtained from the first standard stimulus after the target was significantly larger on the fronto-central and parietal regions compared to that of the sP3 obtained from the last standard stimulus before the target, demonstrating an amplitude increase in the response with the resumption of the original standard stimulus (dishabituation). However, contrary to expectations, the same behavior was not found for SCRs, whose amplitude remained unchanged across the target.

The fact that SCRs evoked by standards were proportionally fewer and smaller compared to those evoked by targets should be attributed to the irrelevance of standards for the task completion. On the contrary, the sP3 from the first standard after a target would seem to occur in correspondence to a local deviance, i.e., at the target-standard change. In this case, in the presence of a contingent change, in some ways unexpected (since task goals were, for the response, the recognition of the standard-deviant change; for the non-response, the recognition of the standard-standard invariant), low-level activation of cortical alertness phenomena could have occurred (i.e., a passive, transient, bottom-up activation of the generalized attention), but not large enough to give rise to an amplitude increase in the SCR. Thus, on the one hand, the local change could be said to reach sufficient magnitude to be detected by the cognitive system (note that this does not presuppose awareness). On the other hand, the resulting level of neuronal activation would be subthreshold to trigger an SCR. This is in line with the assumption that SCRs would be triggered according to a threshold-dependent mechanism [28].

### 4.4. Limitations

The oddball paradigm is not a paradigm specifically designed to study habituation. However, our stated goal was precisely to study habituation phenomena, which are intrinsic to one of the most used paradigms in the field of experimental and clinical psychophysiology, so as to be able to take them into due consideration in future studies.

The present study was based on the average of variables calculated for each stimulation block and not on the evaluation of single trials. This is because we chose to focus on a favorable signal-to-noise ratio to obtain reliable ERP waveforms that allowed us the best possible identification of components. This, however, may have caused some loss of resolution in defining the precise trend of habituation phenomena.

It is important to recognize that several factors, not all easily controllable or accounted for in this study, can influence central and autonomic dynamics. Such factors may include stress, sleep quality, and hormonal fluctuations [53,54]. A specific instance is the menstrual cycle, during which some studies have observed differences in the amplitude and latency of ERP components [54] (e.g., N1, P2, N2, P3) and in the amplitude of evoked SCRs [55,56]. However, variability in study designs and task choices has led to contrasting results in the literature, underscoring the need for more comprehensive investigations [57,58,59,60,61,62]. To address these limitations, our statistical analyses followed a within-subject design, which helps mitigate potential confounding effects of the menstrual cycle on the observed results. Additionally, the analysis of evoked SCRs was based on the cvxEDA algorithm. This method estimates the underlying sympathetic nervous activity, reducing the impact of confounding factors and inter-subject variability, thereby enhancing the reliability of our results.

## 5. Conclusions

In the present work, it was possible to evaluate the phenomena of habituation, recovery, and dishabituation, triggered by stimuli presentation modes in an active (i.e., target count) two-stimulus oddball paradigm. This allowed us to attribute to the target stimulus the emotional-affective connotation of deviant and task-relevant stimulus, which could trigger a more marked OR compared to the standard, task-irrelevant stimulus. Given the paradigm we employed, the PCA-derived lP3a and P3b components of the LPC showed a similar behavior to that of SCRs, and were proposed as a central index of OR. Moreover, the sP3 showed a significant dishabituation phenomenon after the stimulus change. Yet, it did not reach a sufficient level to trigger an SCR/OR, possibly due to the irrelevance of this event in the context of the task.

Considering that the oddball paradigm goals (i.e., discrimination and response) are focused on the target stimulus, it is reasonable to think that the cognitive connotation associated with targets prevails over the emotional-affective one. On the contrary, for the standard task-irrelevant stimulus, both the emotional and cognitive values are minimal at the global level, even if at the local level the cognitive value (i.e., generalized attention) possibly acquires a certain relevance when the standard stimulus signals a deviance from the preceding target (change detection).

## Figures and Tables

**Figure 1 sensors-24-05053-f001:**
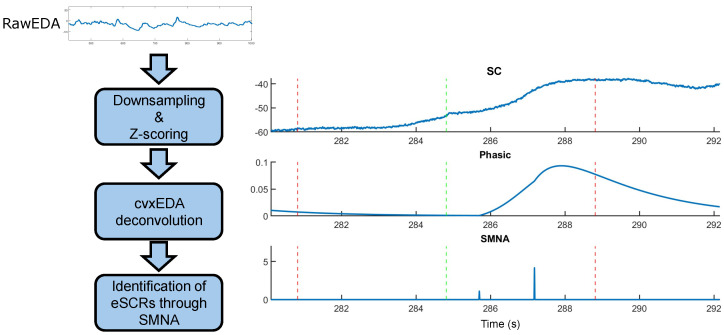
Schematic illustration of the procedure to identify stimulus–evoked SCRs (eSCRs). The raw SC is downsampled at the sampling frequency of 50 Hz and then the z–score is transformed in order to be deconvolved with cvxEDA to obtain an estimate of the phasic component and sudomotor nerve activity (SMNA). The right part of the figure shows an exemplary comparison between raw SC, phasic component, and SMNA to standard (red–dashed vertical lines), and target (green–dashed vertical lines) auditory stimuli. Nonzero SMNA bursts occurring in the 1–5 s interval after the target stimulus onset indicate that the observed SCR is related to the target stimulus itself, whereas no eSCR is present due to the standard stimuli before and after the target.

**Figure 2 sensors-24-05053-f002:**
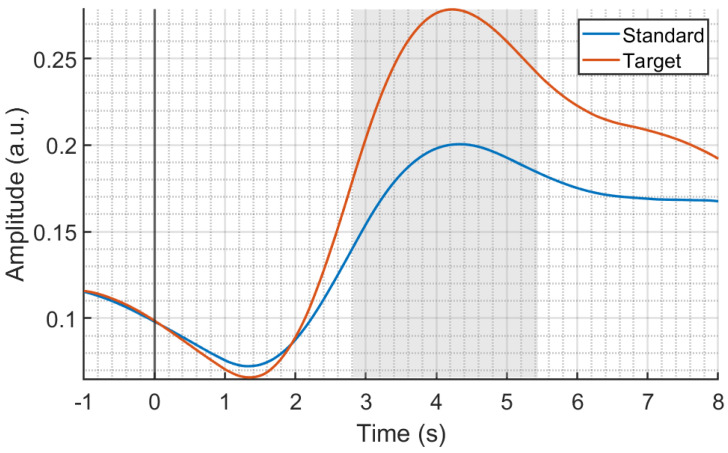
The average SCR responses to the standard (blue) and target (red) stimuli in the −1–8 s interval with respect to the stimulus onset (black vertical line). The gray shaded area indicates a significant difference (*p* < 0.05) between SCRs, with a higher response to targets in the 2.8, 5.4 s interval with respect to the response to standards.

**Figure 3 sensors-24-05053-f003:**
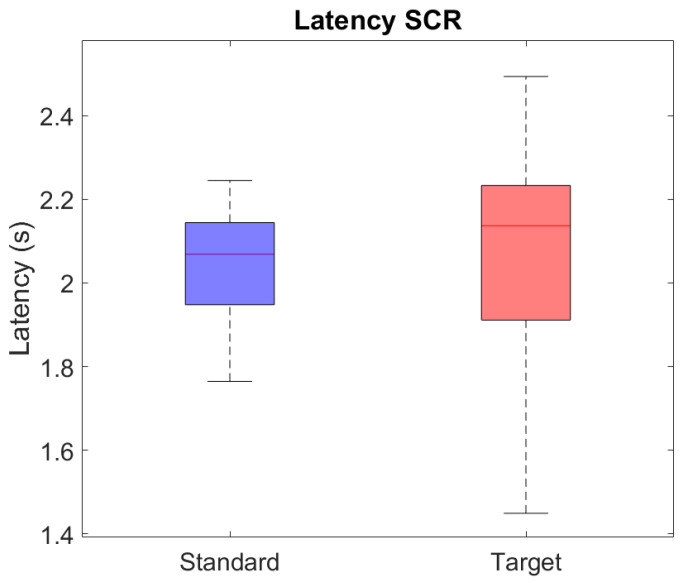
Results of the statistical analysis on the latency of sudomotor nerve activity (SMNA) responses to standard (blue) and target (red) stimuli (median ± mean absolute error (mae); No significant difference in the latency of responses was present.

**Figure 4 sensors-24-05053-f004:**
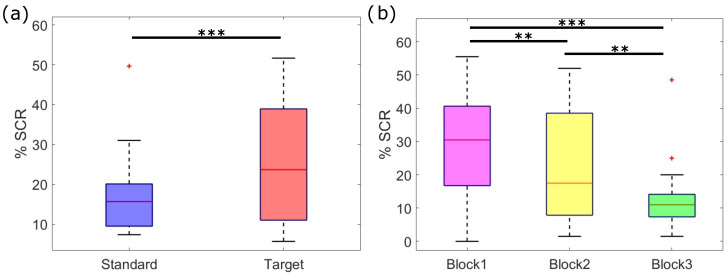
Statistical analysis of the percentage of stimulus-evoked SCRs (**: *p* < 0.01, ***: *p* < 0.001). Red plus signs indicate outliers in the distribution. (**a**) Percentage of SCRs to standard (blue) and target (red) stimuli, irrespective of the stimulation block. Target stimuli elicited more SCRs compared to standard stimuli (SCRtarget = 32%, SCRstandard = 16%, *p* < 0.001); (**b**) percentage of SCRs evoked by any stimulus type across blocks. There is a significant decrease in the percentage of SCRs over time.

**Figure 5 sensors-24-05053-f005:**
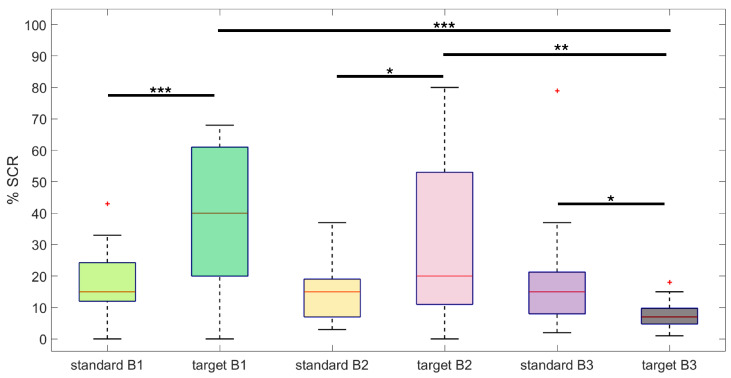
Post hoc analysis of the interaction between stimulus type (standard, target) and stimulation block (B1, B2, B3) on the percentage of evoked SCRs (*: *p* < 0.05, **: *p* < 0.01, ***: *p* < 0.001). Red plus signs indicate outliers in the distribution. There is a significant habituation to the target stimuli, as the percentage of SCRs evoked by targets decreases over blocks. On the other hand, the percentage of SCRs evoked by standard stimuli does not differ across stimulation blocks. Interestingly, the number of responses in the third stimulation block is higher for standard stimuli, compared to targets.

**Figure 6 sensors-24-05053-f006:**
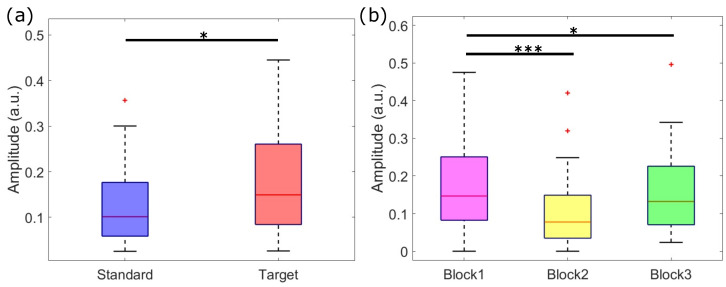
Statistical analysis of stimulus–evoked SCRs’ amplitude (*: *p* < 0.05, ***: *p* < 0.001). (**a**) Mean ± standard error of the amplitude in response to standard (blue) and target (red) stimuli. SCRs to target stimuli had a significantly higher amplitude with respect to SCRs to standards; (**b**) mean ± standard error of SCRs’ amplitude over stimulation blocks, irrespective of the stimulus type. Amplitude in the first block was higher than the amplitude in the second and third blocks, but no significant difference was present between the second and third blocks.

**Figure 7 sensors-24-05053-f007:**
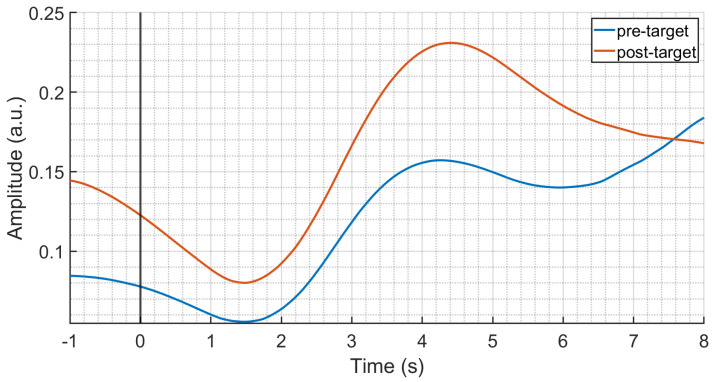
Grand–average SCR responses to the standard stimuli presented immediately before (pre–target, blue) and after (post–target, red) target stimuli in the −1–8 s interval with respect to the stimulus onset (black vertical line). No significant difference has been found between responses.

**Figure 8 sensors-24-05053-f008:**
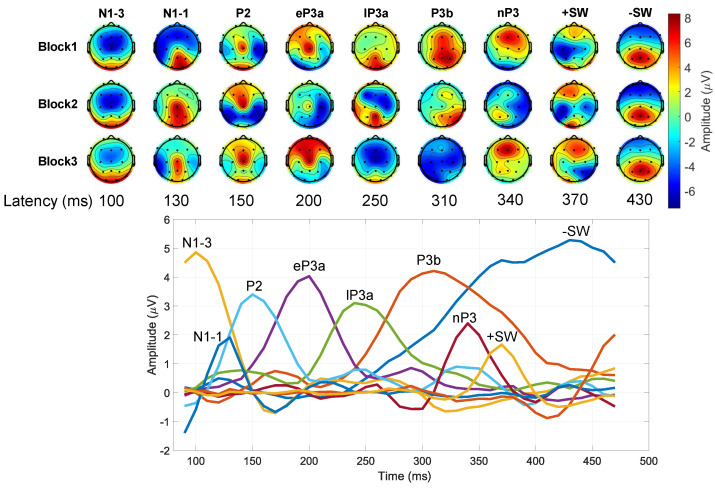
Results of the temporal PCA decomposition on subject–average ERPs. Top: topographical grand-average voltage distribution (i.e., scores) of each identified factor across the three stimulation blocks. Bottom: time–course of each factor (i.e., loadings) in the 90–470 ms range, and its peak latency in msec.

**Figure 9 sensors-24-05053-f009:**
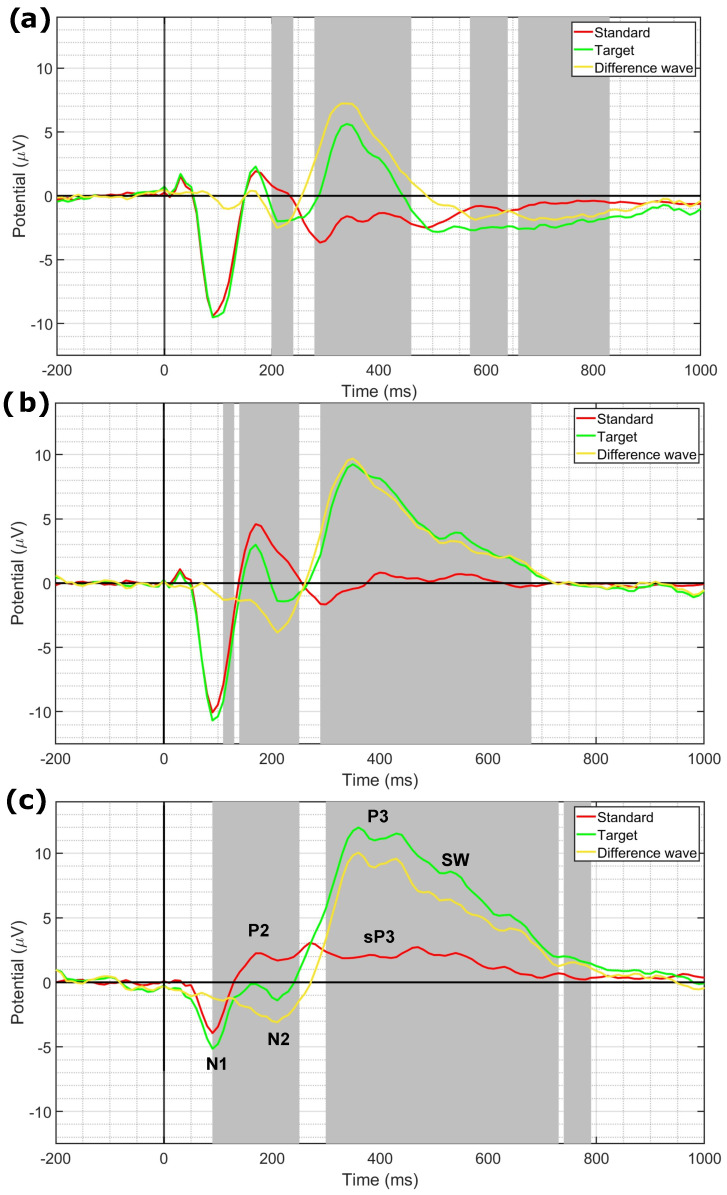
Grand-average ERP responses to the standard (red) and target (green) stimuli, and the difference between them (difference wave; yellow), evaluated at (**a**) Fz, (**b**) Cz, and (**c**) Pz channels. Responses are plotted in the (−200–1000) ms interval with respect to the stimulus onset (black vertical line). The gray shaded areas indicate significant differences between responses to standard and target stimuli (*p* < 0.05).

**Figure 10 sensors-24-05053-f010:**
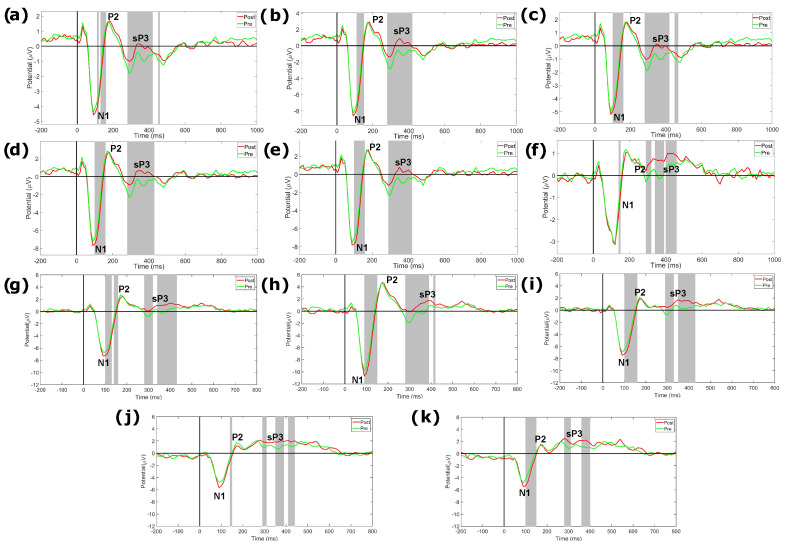
Results of the statistical comparison between ERP responses to standard stimuli observed immediately before (pre–target) and after (post–target) target stimuli. We report only those channels that showed a significant difference between responses, including (**a**) Fp1, (**b**) Fz, (**c**) Fp2, (**d**) F3, (**e**) F4, (**f**) T3, (**g**) C3, (**h**) Cz, (**i**) C4, (**j**) P3, and (**k**) P4. For each of them, we show the grand-average ERPs associated with the pre–target and post–target conditions, respectively, in the −200, 1000 ms time range. Gray shaded areas indicate a significant difference between ERP amplitudes (*p* < 0.05). Post-target responses showed a more negative N1 amplitude and a more positive standard P3 (sP3) amplitude, with respect to pre-target responses.

**Figure 11 sensors-24-05053-f011:**
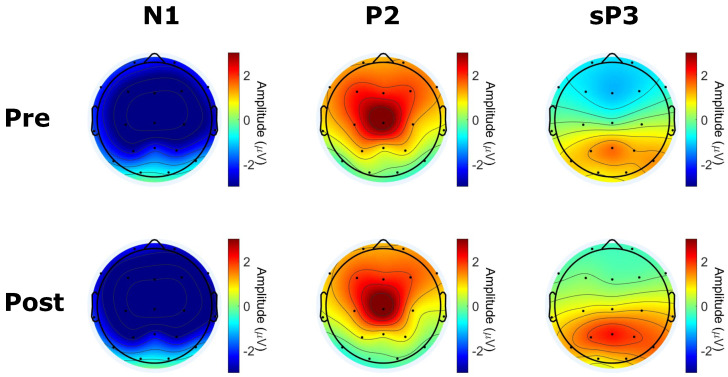
Grand-average scalp map distribution of individual ERP components’ amplitude (N1, P2, standard P3 (sP3)) associated with the standard stimuli presented before (pre) and after (post) target stimuli. Amplitude was calculated as the average within the a priori defined time range for each component. Post responses showed a significantly more negative N1 amplitude and a more positive sP3 amplitude, with respect to pre–responses.

**Table 1 sensors-24-05053-t001:** Mean ± standard error of the LPC components factor scores (i.e., early P3a (eP3a), late P3a (lP3a), P3b, nP3, positive SW (+SW), and negative SW (−SW)) across stimulation blocks. Significant differences between blocks are highlighted in bold, together with the associated *p*-value.

	Block1	Block2	Block3	Sig
eP3a Fz	0.120 ± 0.274	−0.102 ± 0.242	0.290 ± 0.254	*p* > 0.05
eP3a Cz	0.368 ± 0.343	0.125 ± 0.241	0.326 ± 0.260	*p* > 0.05
**lP3a Cz**	**0.090 ± 0.303**	−0.076 ± 0.304	**−0.659 ± 0.323**	***p*** **< 0.05**
lP3a Pz	0.335 ± 0.170	0.275 ± 0.252	−0.057 ± 0.248	*p* > 0.05
**P3b Pz**	**0.529 ± 0.231**	0.219 ± 0.265	**−0.229 ± 0.244**	***p*** **< 0.05**
nP3 Fz	0.531 ± 0.260	−0.012 ± 0.196	0.512 ± 0.250	*p* > 0.05
+SW Fz	0.137 ± 0.183	0.171 ± 0.236	0.218 ± 0.289	*p* > 0.05
−SW Fz	−0.607 ± 0.146	−0.700 ± 0.147	−0.496 ± 0.134	*p* > 0.05

## Data Availability

The raw data presented in this study are available on request from the corresponding author.

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
