# Peer review of "Habituation of Central and Electrodermal Responses to an Auditory Two-Stimulus Oddball Paradigm"

_sensors, 2024, doi:10.3390/s24155053_

Round 1
Reviewer 1 Report
Comments and Suggestions for Authors
This is an interesting paper examining SCR and P300 habituation in an auditory oddball task.
Unfortunately, the view of the P300 as including only P3a and P3b is outdated and indefensible. Some of the literature negating that view is cited but not integrated into the MS. One citation of importance in this regard is listed in the references but not mentioned in text.
With SCR, it is not clear on what basis the proximal rather than distal sites were chosen to record from. There is no mention of the electrode paste used or the exciting voltage.
With EEG, there is no mention of the number of accepted epochs.
They state on P 5. L211 that they used SOAs between 5 and 6 s; at L 239 it states “between 4s and 5s”.
Stimulus frequencies are stated as 10kHz for standards and 15kHz for targets. These are rather higher than is common, and there is no stated check on their audibility. They were not counterbalanced, so that any effect claimed as S/T is confounded with frequency differences (and potential reduced hearing level). I did not see a mention of stimulus duration, rise/fall times, or intensity.
The number of retained epochs for analysis is not apparent.
Fig. 8 legend refers to blue and red lines, but the figure has no blue lines.
Fig. 9: “P3a” topography is atypical.
Discussion is verbose and many statements are unsupported, reflecting the misuse of P3a/P3b components and their outdated understanding of the P300 family.
Author Response
|
Comments 1: This is an interesting paper examining SCR and P300 habituation in an auditory oddball task. Unfortunately, the view of the P300 as including only P3a and P3b is outdated and indefensible. Some of the literature negating that view is cited but not integrated into the MS. One citation of importance in this regard is listed in the references but not mentioned in text.
|
|
Response 1: We thank the reviewer for his/her comment. We recognize that there are consistent findings negating the view of P300 as a single entity, and that such component is instead described as a Late Positive Complex (LPC), which includes P3a, P3b, and the novelty P3 (nP3). Often, these components are followed by a late and broadly distributed Slow Wave (SW) component (1). In our original analysis, we attempted to identify such components by means of visual inspection on the time course of grand-average ERPs and their associated scalp map distribution, and the guidance from previous literature. However, we were able to find only P3a and P3b components, but neither nP3 nor SW could be identified. This may be related to the temporal overlapping of LPC components, which hinders their individual contribution to the observed responses (2).
To address the reviewer’s concern, we reviewed our analysis by applying the method of temporal PCA to isolate and identify the contribution of individual LPC components and investigate habituation to target stimuli. Temporal PCA decomposes ERP responses into a set of orthogonal (i.e., uncorrelated) components, whose temporal course and spatial distribution are described by factor loadings and factor scores, respectively. Specifically, factor loadings represent the time course of each component, and are fixed across channels, conditions and subjects. Instead, factor scores quantify the contribution of each component to the amplitude of each response (2). Operationally, we concatenated average ERPs to target stimuli in the (90-470)ms time range across subjects and for each of the three stimulation blocks, and we applied temporal PCA decomposition through the erpPCA functions implemented by Kayser J and Tenke CE (http://psychophysiology.cpmc. columbia.edu/software/) (3). The input data consisted of 1140 trials (i.e., channels × blocks × subjects) by 39 time points. The covariance matrix underwent Kaiser normalization, and factors were rotated through unrestricted Varimax rotation. Following factors’ extraction order, they were associated to established components through visual inspection of their peak latency, polarity and scalp map. Finally, for each component contributing the LPC, we extracted the corresponding peak’s amplitude factor scores for each subject and for each stimulation block. Particularly, we selected channels of major interest based on the maximums of the scalp topography (i.e., factor scores) associated to each component (2).
We report the results of temporal PCA analysis in Figure 1. In particular, we show the scalp topography of factor scores for each identified component and across stimulation blocks, together with time course of each factor (i.e., loadings) and their peak latency. The first 9 factors accounted for about the 94.8% of the total variance, with a maximum of 38.7% for the first factor and a minimum of 1.15% for the ninth factor. We did not consider any further factor since they explained a variance lower than 1%. We found two subcomponents of the N1, i.e., N1-3 and the N1-1, at a latency of 100ms and 130ms respectively (1,4), followed by a P2 component peaking at 150ms and centered at the vertex Cz. Concerning the LPC, we found both an early P3a (eP3a) and a late P3a (lP3a), peaking at 200ms and 250ms, respectively (5). The eP3a was distributed towards central and frontal regions around Cz and Fz, whereas lP3a was shifted towards central and parietal regions around Cz and Pz. Furthermore, we observed a P3b component with latency of 310ms and a topography focused around Pz, followed by a frontal nP3 peaking at 340ms around Fz (1,5). Finally, we found two subcomponents of the SW, namely the positive SW (+SW) at 370ms and the negative SW (-SW) at 430ms. The latter subdivision of SW is in line with (5,6). Both +SW and -SW topographies showed a maximum and a minimum, respectively, in the frontal region around Fz.
Figure 1 Results of the temporal PCA decomposition on subject-average ERPs. Top: topographical grand-average voltage distribution (i.e., scores) of each identified factor across the three stimulation blocks. Bottom: time-course of each factor (i.e., loadings) in the (90-470)ms range, and its peak latency in ms.
We tested for the occurrence of habituation to target stimuli over stimulation blocks through a permutation-based one-way repeated-measures ANOVA (10000 permutations, α=0.05) on the factor scores of each LPC component, with the three stimulation blocks as the main effect of interest. Multiple comparisons were controlled with the false discovery rate (FDR) method. We conducted post-hoc comparisons through multiple paired t-tests, and we adjusted p-values with the Bonferroni correction. We observed a significant effect for the stimulation blocks on the amplitude of lP3a component at Cz (F2,40 = 4.17, p<0.05), and for the amplitude of P3b component at Pz (F2,40 = 5.02, p<0.05). Post-hoc analysis highlighted a decrease in the amplitude of responses to target stimuli over time, with both lP3a and P3b showing a higher amplitude in the first stimulation block, with respect to the third. As depicted in Figure 1, the grand-average scalp distribution of -SW showed a similar trend at Fz, but without any statistical significance. Likewise, we did not find any significant effect for the stimulation blocks on the factor scores of eP3a, nP3, and +SW.
We added a Temporal PCA subsection to the Materials and Methods section, which we report here for the ease of the reviewer: “We applied temporal PCA to investigate the occurrence of habituation to target stimuli on individual ERP components. Temporal PCA decomposes ERP responses into a set of orthogonal (i.e., uncorrelated) components, whose temporal course and spatial distribution are described by factor loadings and factor scores, respectively. Factor loadings represent factors’ (i.e., components) time course, and are fixed across channels, conditions and subjects. Instead, factor scores represent how much a factor contributes to the amplitude (i.e., voltage) of each observation [35]. Operationally, we further epoched subject-average ERPs to target stimuli in the (90-470)ms time range for each stimulation block, and we applied temporal PCA decomposition through the erpPCA functions implemented by Kayser J and Tenke CE (http://psychophysiology.cpmc. columbia.edu/software/) [36]. The input data matrix consisted of 1140 trials (i.e., channels × blocks × subjects) by 39 time points. Prior to svd decomposition, the covariance matrix underwent Kaiser normalization. Factors were rotated through unrestricted Varimax rotation. Following their extraction order, we identified factors in terms of their peak latency, polarity and scalp map. Finally, for each component contributing the LPC, we extracted the corresponding peak’s amplitude factor scores for each subject and for each stimulation block. Particularly, we selected channels of major interest based on the maximums of the scalp topography (i.e., factor scores) associated to each component [35].”
We also changed the EEG statistical analysis subsection as follows: “We then investigated for the occurrence of habituation to target stimuli over time. To this aim, we performed a permutation-based one-way repeated-measures ANOVA (10000 permutations, α=0.05, corrected with FDR) on the factor scores of each LPC component identified through temporal PCA (see Section 2.4), with the three stimulation blocks as the effect of interest. We conducted post-hoc comparisons through multiple paired t-tests, and we adjusted p-values with the Bonferroni correction.”
Results were updated to account for the new findings following temporal PCA decomposition and habituation investigation on PCA components. We report here the edited section for the ease of the reviewer: “The results of habituation analysis on LPC components’ amplitude (i.e., factor scores) over stimulation blocks are resumed in Table 1. We observed a significant effect for the stimulation blocks on the amplitude of lP3a component at Cz (F2,40 = 4.17, p<0.05), and for the amplitude of P3b component at Pz (F2,40 = 5.02, p<0.05). Post-hoc analysis highlighted a decrease in the amplitude of responses to target stimuli over time, with both lP3a and P3b showing a higher amplitude in the first stimulation block, with respect to the third. As depicted in Fig.9, the grand-average scalp distribution of -SW showed a similar trend at Fz, but without any statistical significance. Likewise, we did not find any significant effect for the stimulation blocks on the factor scores of eP3a, nP3, and +SW.”
|
|
Comments 2: With SCR, it is not clear on what basis the proximal rather than distal sites were chosen to record from. There is no mention of the electrode paste used or the exciting voltage. Response 2: We thank the reviewer for the comment. We have chosen proximal sites for SCR recording as this site has been reported to be less susceptible to movement artifacts and provide a wider area for electrodes fixing compared to distal sites (7). Concerning the electrode paste, we have used pre-gelled Ag/AgCl electrodes. The applied exciting voltage was 0.5V. “EEG and SC were acquired synchronously using a DSI-24 system from Wearable Sensing at the sampling frequency of 300Hz. The headset was composed of 19 dry electrodes placed according with the 10-20 international system. Electrode impedance was always kept below 1MΩ in accordance with the manufacturer guidelines. All channels were referenced to Pz. SC was recorded via a dedicated auxiliary channel in the EEG cap applying an exciting voltage of 0.5V. A pair of pre-gelled Ag/AgCl electrodes were placed on the proximal phalanx of the first and second fingers of the non-dominant hand, respectively. Proximal sites were chosen to minimize the presence of movement artifacts in the recordings [30].”
|
|
Comments 3: With EEG, there is no mention of the number of accepted epochs. Response 3: We thank the reviewer for the comment. We believe the reviewer is referring to the number of clean epochs retained after EEG preprocessing. In this regard, we retained an average of 295 epochs (min: 280; max: 300) for the standard stimuli, and an average of 74 epochs (min: 70; max: 75) for the target stimuli.
|
|
Comments 4: They state on P 5. L211 that they used SOAs between 5 and 6 s; at L 239 it states “between 4s and 5s”.
Comments 5: Stimulus frequencies are stated as 10kHz for standards and 15kHz for targets. These are rather higher than is common, and there is no stated check on their audibility. They were not counterbalanced, so that any effect claimed as S/T is confounded with frequency differences (and potential reduced hearing level). I did not see a mention of stimulus duration, rise/fall times, or intensity.
Comments 6: The number of retained epochs for analysis is not apparent.
Comments 7: Fig. 8 legend refers to blue and red lines, but the figure has no blue lines. difference between them (difference wave; yellow), evaluated at a) Fz, b) Cz, and c) Pz channels. Responses are plotted in the (-200, 1000)ms interval with respect to the stimulus’ onset (black vertical line). The gray shaded areas indicate significant differences between responses to standard and target stimuli (p<0.05).”
Comments 8: Fig. 9: “P3a” topography is atypical.
Comments 9: Discussion is verbose and many statements are unsupported, reflecting the misuse of P3a/P3b components and their outdated understanding of the P300 family.
|
|
|
|
|
|
|
|
|
|
|
References
- Barry RJ, Steiner GZ, De Blasio FM, Fogarty JS, Karamacoska D, MacDonald B. Components in the P300: Don’t forget the Novelty P3! Psychophysiology. 2020;57(7):e13371.
- Scharf F, Widmann A, Bonmassar C, Wetzel N. A tutorial on the use of temporal principal component analysis in developmental ERP research – Opportunities and challenges. Dev Cogn Neurosci. 2022 Apr 1;54:101072.
- Kayser J, Tenke CE. Optimizing PCA methodology for ERP component identification and measurement: theoretical rationale and empirical evaluation. Clin Neurophysiol. 2003 Dec 1;114(12):2307–25.
- Näätänen R, Picton T. The N1 Wave of the Human Electric and Magnetic Response to Sound: A Review and an Analysis of the Component Structure. Psychophysiology. 1987;24(4):375–425.
- Barry RJ, Steiner GZ, De Blasio FM. Reinstating the Novelty P3. Sci Rep. 2016 Aug 11;6(1):31200.
- Loveless N e., Simpson M, Näätänen R. Frontal Negative and Parietal Positive Components of the Slow Wave Dissociated. Psychophysiology. 1987;24(3):340–5.
- Boucsein W. Electrodermal Activity. Springer Science & Business Media; 2012. 635 p.
Reviewer 2 Report
Comments and Suggestions for Authors
When almost half of the participants are women of reproductive age, the change in autonomic and central nervous system parameters (ERP and SCR) during the menstrual hormonal cycle must be taken into account.
Therefore, the females results should be reanalyzed according to the phases of the ovarian-hormonal cycle and re-presented in the improved paper.
here are additional comments:
1. What is the main question addressed by the research?
The aim of the present work was to unveil any correspondence between SCRs (as typical peripheral OR measures) and ERPs responses (as potential central OR candidates) by studying habituation, recovery, and dishabituation during a traditional two-stimulus auditory odd-ball paradigm.
2. a) What parts do you consider original or relevant for the field?
The originality that authors made to the standard oddball paradigm was to use longer stimulus-onset asynchronies than usual, to allow them the resolution of the individual SCRs without causing their temporal summation.
2 b) What specific gap in the field does the paper address?
A number of studies have shown that physiologic hormonal fluctuations in estrogen and progesterone during the menstrual cycle can affect central and autonomic regulation of psychophysiological functions, including the ERP (Fleck, K. M., & Polich, J. (1988). P300 and the menstrual cycle. Electroencephalography and clinical neurophysiology, 71(2), 157–160. https://doi.org/10.1016/0168-5597(88)90076-7), SCR (Gómez-Amor, J., Martínez-Selva, J. M., Román, F., & Zamora, S. (1990). Electrodermal activity in menstrual cycle phases: a comparison of within- and between-subjects designs. International journal of psychophysiology : official journal of the International Organization of Psychophysiology, 9(1), 39–47. https://doi.org/10.1016/0167-8760(90)90006-y) orienting reactions and attention (Ronca, F., Blodgett, J. M., Bruinvels, G., Lowery, M., Raviraj, M., Sandhar, G., Symeonides, N., Jones, C., Loosemore, M., & Burgess, P. W. (2024). Attentional, anticipatory and spatial cognition fluctuate throughout the menstrual cycle: Potential implications for female sport. Neuropsychologia, 108909. Advance online publication. https://doi.org/10.1016/j.neuropsychologia.2024.108909
4. What specific improvements should the authors consider regarding the methodology? What further controls should be considered?
When almost half of the participants are women of reproductive age, the change in autonomic and central nervous system parameters (ERP and SCR) during the menstrual hormonal cycle must be taken into account.
Therefore, the female’s results should be reanalyzed according to the phases of the ovarian-hormonal cycle and re-presented in the improved paper
5. In this way the conclusions could change..
6. References may be appropriate if the important issues listed above are addressed and cited
I am ready to review this manuscript again after the corrections have been made
instead «Twenty healthy volunteers (age 24 ± 4, 10 females) participated TO the study» it would be more correct to write “Twenty healthy volunteers (age 24 ± 4 years, 10 women) participated IN the study”.
Author Response
|
Comments 1: A number of studies have shown that physiologic hormonal fluctuations in estrogen and progesterone during the menstrual cycle can affect central and autonomic regulation of psychophysiological functions, including the ERP (Fleck, K. M., & Polich, J. (1988). P300 and the menstrual cycle. Electroencephalography and clinical neurophysiology, 71(2), 157–160. https://doi.org/10.1016/0168-5597(88)90076-7), SCR (Gómez-Amor, J., Martínez-Selva, J. M., Román, F., & Zamora, S. (1990). Electrodermal activity in menstrual cycle phases: a comparison of within- and between-subjects designs. International journal of psychophysiology : official journal of the International Organization of Psychophysiology, 9(1), 39–47. https://doi.org/10.1016/0167-8760(90)90006-y) orienting reactions and attention (Ronca, F., Blodgett, J. M., Bruinvels, G., Lowery, M., Raviraj, M., Sandhar, G., Symeonides, N., Jones, C., Loosemore, M., & Burgess, P. W. (2024). Attentional, anticipatory and spatial cognition fluctuate throughout the menstrual cycle: Potential implications for female sport. Neuropsychologia, 108909. Advance online publication. https://doi.org/10.1016/j.neuropsychologia.2024.108909 Response 1: We thank the reviewer for the comment. In the literature there are contrasting results regarding a potential effect of the menstrual cycle on ERP responses. While some authors found an effect for the specific cycle period on the amplitude of N1 and P2 components (1), others did not observe any consistent change over the course of the menstrual cycle (2,3). Particularly, in the auditory discriminant paradigm cited by the reviewer (2), it has been reported no difference in either the amplitude or the latency for any of the key components involved in the task (i.e., N1, P2, N2, and P3). In this light, we believe that further investigations should be necessary to confirm an effect of the menstrual cycle period on ERP and SCR responses. “It is important to recognize that several factors, not all easily controllable or accounted for in this study, can influence central and autonomic dynamics. Such factors may include stress, sleep quality, and hormonal fluctuations [55,56]. A specific instance is the menstrual cycle, during which some studies have observed differences in the amplitude and latency of ERP components [56] (e.g., N1, P2, N2, P3) and in the amplitude of evoked SCRs [57,58]. However, variability in study designs and task choices has led to contrasting results in the literature, underscoring the need for more comprehensive investigations [59,60]. To address these limitations, our statistical analyses followed a within-subject design, which helps mitigate potential confounding effects of the menstrual cycle on the observed results. Additionally, the analysis of evoked SCRs was based on the cvxEDA algorithm. This method estimates the underlying sympathetic nervous activity, reducing the impact of confounding factors and inter-subject variability, thereby enhancing the reliability of our results.”
Comments 2: When almost half of the participants are women of reproductive age, the change in autonomic and central nervous system parameters (ERP and SCR) during the menstrual hormonal cycle must be taken into account. Response 2: We thank the reviewer for the comment. We amended the lack of control on the potential confounding effect of the menstrual cycle period on the observed responses in Section 4.4 Limitations. Please see Response 1 for a detailed answer to the concern.
Comments 3: In this way the conclusions could change. Response 3: We extended Section 4.4 Limitations of the manuscript (please see Response 1).
Comments 4: References may be appropriate if the important issues listed above are addressed and cited. Response 4: We have addressed the reviewer’s concern stated at Comment 1 by updating the Limitations subsection of the manuscript together with the addition of references to relevant state-of-the-art research.
|
|
4. Response to Comments on the Quality of English Language |
|
Point 1: instead «Twenty healthy volunteers (age 24 ± 4, 10 females) participated TO the study» it would be more correct to write “Twenty healthy volunteers (age 24 ± 4 years, 10 women) participated IN the study”. |
|
Response 1: We thank the reviewer for the comment. We have updated the sentence according to the reviewer’s suggestion.
|
Round 2
Reviewer 1 Report
Comments and Suggestions for Authors
A substantial improvement!
Comments on the Quality of English LanguageSome of the expressions could be improved by a native English speaker, but these do not prevent communication.
Author Response
Comment 1: "A substantial improvement!"
Response 1: "We thank the reviewer for his/her comment"
Comments on the Quality of English Language
Comment 1: "Some of the expressions could be improved by a native English speaker, but these do not prevent communication."
Response 1: "We thank the reviewer for his/her comment. We revised the manuscript to improve the quality of English language."
Reviewer 2 Report
Comments and Suggestions for Authors
I'm satisfied with improved made
Author Response
Comment 1: "I'm satisfied with improved made"
Response 1: "We thank the reviewer for his/her comment"